# A Retrospective Study on Evacuation and Long-Term Displacement Among Home-Visit Psychiatric Nursing Service Users in the Aftermath of the 2024 Noto Peninsula Earthquake

**DOI:** 10.3390/ijerph22111688

**Published:** 2025-11-07

**Authors:** Hisao Nakai, Masato Oe, Yutaka Nagayama, Shingo Oe, Mayu Tokuoka, Chinatsu Yamaguchi, Koji Tanaka

**Affiliations:** 1Faculty of Nursing, University of Kochi, Ike, Kochi 781-8515, Kochi, Japan; 2School of Nursing, Kanazawa Medical University, Kahoku 920-0293, Ishikawa, Japan; oemasato@kanazawa-med.ac.jp (M.O.);; 3Nursing Department, Kanazawa Medical University Hospital, Uchinada, Kahoku 920-0293, Ishikawa, Japan; 4Faculty of Nursing, Ishikawa Prefectural Nursing University, Gakuendai, Kahoku 929-1210, Ishikawa, Japan; 5Faculty of Health Sciences, Institute of Medical, Pharmaceutical and Health Sciences, Kanazawa University, Kodatsuno, Kanazawa 920-1192, Ishikawa, Japan

**Keywords:** 2024 Noto peninsula earthquake, psychiatric home-visit nursing, long-term displacement, retrospective study

## Abstract

The aim of this retrospective study was to identify the influencing factors of prolonged evacuation among home-visit psychiatric nursing services patients affected by the 2024 Noto Peninsula Earthquake. We examined the associations between demographic factors, mental illness-related factors, living environment factors, and evacuation status. We also visualized evacuation routes using a geographic information system and analyzed their characteristics. We used data from medical records of patients using a single home-visit psychiatric nursing provider in northern Noto, Ishikawa Prefecture, Japan, an area severely affected by the 2024 earthquake. The study population comprised 115 patients with a mean (standard deviation) age of 53.3 (16.8) years; 64 (55.7%) were women and 51 (44.3%) were men. The median (interquartile range) total number of evacuation days was 208 (192–213) days, and the median (interquartile range) length of stay at the initial shelter was 2.0 (2.0–3.0) days. Binomial logistic regression analysis, adjusted for sex and age, showed that factors associated with prolonged evacuation were an initial shelter stay of <23 days (odds ratio: 3.26, 95% confidence interval: 1.15–9.26; *p* = 0.026) and the initial shelter being a public shelter (odds ratio: 4.56, 95% confidence interval: 1.49–13.95; *p* = 0.008). Geographic information system spatial analysis showed that for the three participants with the highest number of evacuations, the total distance traveled (minutes) for evacuation was 884.1 km (678.9 min), 159.0 km (158.8 min), and 36.8 km (54.8 min). These findings suggest that initial evacuation behaviors and shelter selection may significantly affect evacuation duration among home-visit psychiatric nursing patients.

## 1. Introduction

On 1 January 2024, the magnitude 7.6 Noto Peninsula earthquake (NPE) hit the northern Noto Peninsula in Ishikawa Prefecture, Japan, and the earthquake’s epicenter was in the same area [1]. As of April 2024, Ishikawa Prefecture had a population of approximately 1.1 million; the northern Noto region of the prefecture, which sustained the most severe damage from the NPE, had a population of approximately 160,000 [2].

The NPE resulted in seismic intensities of 6 to 7 in the northern Noto region [3]. Substantial ground uplift owing to crustal deformation considerably altered the coastal topography of this region [4]. Furthermore, a large-scale urban fire erupted in Wajima [5,6], and there was extensive damage to coastal areas from the resulting tsunami [7,8]. The earthquake occurred in winter, exacerbating the difficulties of rescue and disaster relief efforts because of sub-freezing temperatures, cold rain, snow, cut off roads, damaged infrastructure, and the strong impact of the earthquake on the northern peninsula, the most remote area from the mainland [9].

Residents in the northern Noto region, where the most severe shaking occurred, began evacuating immediately after the earthquake. However, the influx of evacuees greatly exceeded the capacity of public shelters, leading to severe overcrowding. Because of this, some evacuees were unable to enter public shelters or had to resort to vehicle evacuation [10,11]. The arrival of medical support teams, including the disaster medical assistance team, was delayed owing to widespread road damage, landslides, and frequent aftershocks [11]. Consequently, primary evacuation centers in the affected areas experienced overcrowding and chaos. In response to this chaotic situation, Ishikawa Prefecture actively promoted wide-area evacuation not only within the prefecture but also to other prefectures [12]. However, the various challenges of this situation led to considerable confusion, including insufficient or inadequate information dissemination to secondary shelters, difficulties in securing transportation, and mismatches between evacuee needs and secondary shelters [13]. To address these issues, Ishikawa Prefecture established a 1.5-level shelter in a large-scale sports facility in Kanazawa, the central city of Ishikawa Prefecture. This was designated as a 1.5-level shelter because it was a temporary emergency shelter for evacuees transitioning from primary (level 1) shelters to secondary (level 2) shelters. The shelter was designed to provide temporary accommodation for individuals with pre-existing conditions, those requiring specific support, or those who found it difficult to live in public shelters until they could be placed in secondary shelters [13]. However, the legal status of the 1.5-level shelters was unclear, and the responsible entities were not clearly defined, leading to difficulties in coordination [13]. Furthermore, the lack of clarity about funding for shelter establishment and operation, shortages in personnel and supplies for support, and a lack of specialized care personnel to look after evacuees posed substantial operational challenges [14]. The living conditions of some evacuees were disrupted, and some were forced to move between multiple shelters. Many of these evacuees had pre-existing conditions requiring specialized care and treatment [15] and included individuals with existing mental health conditions living at home. Some of these individuals were affected by a lack of coordination with medical institutions for continued treatment and shortages of regularly used medications [16].

According to 2022 estimates, approximately 17,900 individuals with mental disorders were receiving outpatient care in Ishikawa Prefecture, of whom approximately 4300 used home-visit psychiatric nursing services (HPNS) [17]. It is well documented that the deinstitutionalization of patients with mental illnesses in Japan has been greatly delayed [18]. As well as increasing social security costs owing to Japan’s declining birthrate and aging population, there has been a recent trend towards community-based care for individuals with mental disorders [19], and the average length of stay in psychiatric hospital beds is on a downward trend [20]. With the shift towards community integration for patients with mental illnesses, there has been an increase in the number of HPNS providers targeting home-based patients [21]. Continuous care through HPNS is effective in preventing readmissions for home-based patients with mental illnesses [22].

Previous research has documented the severe effects of prolonged post-disaster evacuation periods on the physical and mental well-being of evacuees. Specifically, there is an association between disaster exposure and mental health issues, with evidence suggesting an increased prevalence of mental disorders [23,24,25]. Furthermore, many studies have demonstrated that prolonged evacuation can lead to the persistence of mental health problems, decline in physical and mental functions, and isolation owing to separation from familiar community networks [26,27]. Individuals with pre-existing mental illnesses are particularly vulnerable to physical and economic constraints during disaster-induced evacuations [28]. Exposure to natural disasters can exacerbate mental instability and increase the risk of worsening pre-existing mental disorders [25]. An overview of research on the 2024 NPE victims identified numerous studies focusing on the risks of pre-existing conditions and infections among older people because of the severe damage in areas with high aging rates. Specifically, reports indicate worsening nutritional status and increased gastrointestinal symptoms such as vomiting among older people [29], as well as an increased risk of infectious disease outbreaks [30]. In public shelters in Himi, Toyama, which is adjacent to Ishikawa Prefecture, overcrowding, poor ventilation, and substandard sanitary conditions led to outbreaks of COVID-19 and influenza and an increased risk of pneumonia, especially among older people [31]. Furthermore, some cancer patients receiving outpatient chemotherapy had to abandon essential treatments because of poor road conditions [32]. The prolonged evacuation period following the 2024 NPE highlighted the need for better management of chronic diseases and provision of care, and more mental health support [33]. However, the evacuation situation and the effect of long-term evacuation on HPNS patients living at home who were forced to move between shelters after temporary evacuation in the 2024 NPE remain to be clarified.

The study aim was to clarify the factors influencing long-term evacuation among patients using HPNS who were affected by the 2024 NPE. Specifically, we examined the associations between long-term evacuation and demographic factors (sex, age), mental illness-related factors (main diseases), living environment factors (municipality of residence before the earthquake, living situation before the earthquake), and evacuation status (e.g., number of evacuations, municipality of the initial shelter, whether the initial shelter was a public shelter). Furthermore, using a geographic information system (GIS), we visualized the evacuation routes taken by HPNS patients and analyzed route characteristics such as distance and time. Clarifying the evacuation status and factors related to long-term evacuation could help to develop evacuation support measures tailored to the circumstances of HPNS patients living at home. The visualization and analysis of evacuation routes using GIS could help to improve the development of future evacuation plans and disaster countermeasures.

## 2. Materials and Methods

### 2.1. Geographical Characteristics of the Noto Peninsula

Ishikawa Prefecture is located in the central part of Honshu and faces the Sea of Japan. The northern part of Ishikawa Prefecture forms the distinctive Noto Peninsula, a landmass that protrudes into the Sea of Japan. The Noto Peninsula is characterized by its complex coastline, rich natural environment, and diverse landscape [34]. As of 1 April 2024, the population of Ishikawa Prefecture was 1,101,105. Approximately 14.7% of the prefecture’s total population (162,361 people) live in the five northern Noto regions, namely, Nanao, Kashima District, Wajima, Suzu, and Hōsu District [2]. The Noto Satoyama Kaido expressway, which connects Kanazawa and Wajima, is the main route for goods transportation, commuting, tourism, and emergency vehicle transport on the peninsula [35,36]. The location of the Noto Peninsula is shown in Figure 1.

This map depicts the routes between the first and fifth shelters, not including the initial route from the participants’ homes to the first shelter after the disaster. HPNS, home-visit psychiatric nursing services.

### 2.2. Data Collection

This was a retrospective study. Information was obtained retrospectively from the medical records of HPNS providers. The participants were patients of a single HPNS provider located in the northern Noto region of Ishikawa Prefecture. In conducting this study, substantial consideration was given to the protection of patients’ personal information. After anonymizing the HPNS provider and patients, information from the HPNS medical records was collected retrospectively for the period from 1 January 2024 to 31 July 2024. Data were accessed for research purposes on 29 April to 31 July 2024. The authors did not have access to information that could identify individual participants during or after data collection.

### 2.3. Survey Contents

Background information was collected on participants’ sex, age, and main diseases. To assess pre-NPE evacuation status, participants’ municipality of residence and living situation before the earthquake were recorded. The following aspects of participants’ post-earthquake status were also assessed: whether the participant remained at home or evacuated after the earthquake, number of days of evacuation, length of stay at each evacuation shelter, whether the initial shelter was a public shelter, living situation after the earthquake, whether the participant was hospitalized during the initial evacuation period, and municipality of the initial shelter. The municipalities of the first five shelters to which participants had been evacuated were identified to enable GIS visualization of the evacuation routes.

### 2.4. Data Analysis

#### 2.4.1. Analysis of Participant Backgrounds

A total of 115 participants were included in this study. Descriptive statistics were used to summarize the demographic characteristics of participants, including the mean and standard deviation (SD) of age and the sex distribution (percentages). The total number of evacuation days and the number of evacuations were calculated and described using the median and interquartile range (IQR). The distribution of evacuation duration for the first five evacuations was also examined across different shelters. Participants were then categorized into two groups: those who remained in their homes after the disaster (stay-at-home group; *n* = 27) and those who evacuated (evacuation group; *n* = 88). For these two groups, the mean age with SD, sex distribution (percentages), and main diseases (percentages) were calculated and compared. For the evacuation group, the shelter was described using the median and IQR for the following variables: total days of evacuation, duration of stay at each shelter (shelters 1–5), and the total number of evacuations.

#### 2.4.2. Analysis of Long-Term Evacuation and Associated Factors

Data were analyzed for the 88 participants in the evacuation group to explore the factors associated with long-term evacuation. As the median total number of evacuation days was already prolonged at 208 days, we defined long-term evacuation as a duration of 212 days or more (the 75th percentile) to more accurately identify the subgroup experiencing the most extreme duration of displacement: those with 212 days or less (1: <75th percentile value) and those with more than 212 days (2: ≥75th percentile value). Chi-square tests and Fisher’s exact tests were used to examine the association between long-term evacuation and participant attributes (age, sex, main diseases) and evacuation status. Logistic regression analysis was subsequently performed to identify independent factors associated with long-term evacuation. The following variables were included as explanatory variables (all were treated as binary variables): age (1: <65 years; 2: ≥65 years); duration of the first evacuation (1: <23 days; 2: ≥23 days); presence of main diseases (1: other than schizophrenia; 2: schizophrenia); pre-earthquake residence (1: outside Wajima; 2: Wajima); living situation before the earthquake (1: cohabitating; 2: living alone); living situation after the earthquake (1: cohabitating; 2: living alone); type of evacuation shelter (1: non-public shelter; 2: public shelter); hospitalization during the first evacuation (1: no hospitalization; 2: hospitalization); and location of the first evacuation shelter (1: outside Wajima; 2: Wajima; 1: outside Suzu; 2: Suzu; 1: outside Kanazawa; 2: Kanazawa).

To identify factors independently associated with long-term evacuation, a binary logistic regression analysis was performed. The dependent variable was long-term evacuation (defined as above), and sex and age (1: <65 years; 2: ≥65 years) were forced into the model as covariates. Variables that showed statistical significance (*p* < 0.05) in the univariate analysis (chi-square test or Fisher’s exact test) were included as explanatory variables. These included living situation before the earthquake, length of stay at the initial shelter, whether the initial shelter was a public shelter, and the municipality of the initial shelter (Wajima, Kanazawa). Before inclusion in the model, multicollinearity was assessed using the variance inflation factor; values >10 indicate potential multicollinearity. The statistical significance level was set at *p* < 0.05 (two-tailed). All data aggregation and statistical analyses were conducted using SPSS version 29 (IBM Corporation, Armonk, NY, USA).

#### 2.4.3. Visualization of Evacuation Routes

GIS was used to analyze the evacuation routes; for the three participants with the highest number of evacuations, the routes from their first to fifth shelter were visualized on a map. To protect personal information, instead of using the exact location data, public facilities that served as shelters at the time of the disaster and had relatively high numbers of evacuees were arbitrarily selected, plotted on the map, and used for route analysis. Spatial analysis was conducted using ArcGIS Pro 3.2.1 (ESRI, Redlands, CA, USA).

## 3. Ethical Considerations

This retrospective study was approved by the institutional review boards at the authors’ affiliated universities (approval nos. C093 and 24-3). Informed consent was obtained from participants using an opt-out approach. The HPNS staff explained the study using an informed consent document, which was also distributed to the participants. This document informed participants of the study’s purpose and significance, the research methods, the voluntary nature of participation, the anonymity of responses, and the assurance that no individual would be identified from their answers.

## 4. Results

### 4.1. Participant Characteristics

The mean age of the 115 participants was 53.3 years (SD 16.8). The sex distribution was 64 women (55.7%) and 51 men (44.3%). The median total number of evacuation days was 208 (IQR 192–213), and the median number of evacuations was 2 (IQR 2−3). Figure 2 shows the distribution of evacuation durations for the first to fifth evacuations, categorized by the municipality where the evacuation shelters were located.

Following the earthquake, 27 participants (23.5%) remained in their homes, and 88 participants (76.5%) evacuated. The mean age of the stay-at-home group was 51.4 years (SD 17.6), and this group contained 14 women (51.9%) and 13 men (48.1%). The main diseases in this group were schizophrenia (13 participants, 48.4%), depression (3 participants, 11.1%), and intellectual disability (2 participants, 7.4%). Additionally, one participant each (3.7%) had bipolar disorder, attention-deficit hyperactivity disorder, delusional disorder, or adjustment disorder.

### 4.2. Cross-Tabulation of Long-Term Evacuation and Explanatory Variables

The evacuation group comprised 88 participants, with a mean age of 53.3 years (SD 16.8); this group contained 50 women (56.8%) and 38 men (43.2%). The median total number of evacuation days was 208 days (IQR 192–213) and the median number of evacuations was 2 (IQR 2–3). A total of 41 participants (46.6%) exceeded the 75th percentile threshold for long-term evacuation, which was 212 days in this study. Table 1 shows the results of the chi-square or Fisher’s exact tests examining the association between long-term evacuation and basic attributes or evacuation status. The following variables were significantly associated with long-term evacuation: length of stay at the initial shelter less than 23 days (*n* = 27, 65.9%; *p* < 0.001), cohabiting before the earthquake (*n* = 38, 52.1%; *p* = 0.023), the initial shelter being a public shelter (*n* = 32, 62.7%; *p* < 0.001), the municipality of the initial shelter being Wajima (*n* = 20, 60.6%; *p* = 0.041), and the initial shelter being located outside of Kanazawa (*n* = 40, 51.9%; *p* = 0.008).

### 4.3. Factors Associated with Long-Term Evacuation: Results of Logistic Regression Analysis Adjusted for Age and Sex

The binary logistic regression analysis, adjusted for age and sex, identified significant associations between long-term evacuation and the following factors: length of stay at the initial shelter less than 23 days (odds ratio: 3.26, 95% confidence interval: 1.15–9.26; *p* = 0.026) and the initial shelter being a public shelter (odds ratio: 4.56, 95% confidence interval: 1.49–13.95; *p* = 0.008) (Table 2).

### 4.4. Spatial Analysis of Evacuation Routes for Participants with Frequent Evacuations

Among the HPNS patients, three individuals evacuated the most, with five relocations each. Two of these individuals evacuated for 206 days and the third for 216 days. The total distances traveled, as determined by GIS-based spatial analysis, were 884.1 km (678.9 min) for ID49, 159.0 km (158.8 min) for ID107, and 36.8 km (54.8 min) for ID 68. Figure 1 illustrates the spatial distribution of the evacuation routes for these three participants.

## 5. Discussion

This study investigated the evacuation status and factors influencing prolonged evacuation among patients using HPNS who were affected by the 2024 NPE. According to the 2023 Patient Survey in Japan, the largest age group for outpatients with schizophrenia and mood disorders was 40–49 years. The sex distribution in this national survey was approximately 50% for each sex for schizophrenia, and approximately 65% women and 35% men for mood disorders [37]. Considering that the participants in the present study were limited to HPNS patients and that the aging rate in the Oku-Noto region in 2023 was 48.1% [2], it is likely that our participants represent a typical population of individuals with mental disorders living in and receiving care in the community in Japan.

A distinctive feature of the 2024 NPE is the exceptionally prolonged duration of evacuation and the number of relocations, even when compared with those of other recent major earthquakes in Japan. For example, at the peak of the 2016 Kumamoto earthquake, approximately 180,000 people were evacuated [38]. A survey of the evacuees conducted by Kumamoto Prefecture found that 36.6% were evacuated for less than 3 days, 28.8% for 1 week to less than 1 month, and only 12.9% for more than 1 month [38]. At the peak of the 2018 Hokkaido Eastern Iburi earthquake, approximately 16,000 people were evacuated, but this number dropped to approximately 2700 after 6 days, and all evacuation shelters were closed within approximately 2 months [39]. Even considering that our participants were HPNS patients, the fact that the median evacuation duration exceeded 200 days indicates that a large number of 2024 NPE evacuees experienced prolonged evacuation.

The results of the binomial logistic regression analysis suggest that among HPNS patients, those whose initial evacuation period was less than 23 days may have been more likely to experience prolonged subsequent evacuation. Although a short initial stay at an evacuation shelter may seem unrelated to prolonged evacuation, because of the extensive damage in the northern Noto Peninsula, several factors (including poor evacuation shelter conditions, a shortage of shelters, and the geographical vulnerability of the peninsula, with limited detour options owing to disrupted road networks) may have increased the dispersal of evacuees from primary evacuation shelters. In fact, the northern Noto region experienced immediate shortages of evacuation shelters after the disaster, as well as harsh winter conditions and delays in the delivery of essential supplies and humanitarian aid owing to disrupted road networks [40,41]. Consequently, Ishikawa Prefecture actively promoted secondary evacuation to areas outside the disaster zone [5]. In response to this, neighboring municipalities around the Noto Peninsula accepted evacuees and provided public evacuation shelters and accommodation [42]. These combined factors may explain why some HPNS patients, particularly those without major impairments in activities of daily living, may have chosen to leave their initial evacuation sites despite having underlying mental health conditions. Most mental disorders are not visually apparent [43]. A study of individuals with mental disorders residing in group homes showed that approximately half were reluctant to disclose their disabilities to others, even during emergency evacuations [44]. Taking this into account, it is possible that HPNS patients were among those who relocated from their initial shelters to subsequent ones relatively early following the 2024 NPE. Furthermore, an initial stay in a public shelter was associated with prolonged evacuation. This suggests that disparities in social support in the community may have affected evacuation duration. It is possible that individuals lacking options such as evacuation to family, relatives, or group homes outside the affected area were more likely to move between public evacuation shelters, resulting in a longer overall evacuation period. Public evacuation shelters are often designed to provide only the minimum necessary facilities for temporary stays [45]. It is also possible that patients moved between shelters in search of better evacuation conditions, which may have increased the number of evacuations, thus extending the evacuation period.

Three patients using HPNS experienced the highest number of evacuation-related relocations, with five moves each. All three patients were evacuated for over 200 days. Spatial analysis using GIS showed that their total travel distance exceeded approximately 150 km, with the longest distance surpassing 880 km. The geographical characteristics of the Noto Peninsula, which is surrounded by sea and lacks connecting land routes to adjacent municipalities, likely exacerbated the travel distances required for evacuation. This is because overland support from neighboring municipalities was unavailable. Individuals with pre-existing mental illnesses or those experiencing mental health problems are known to be vulnerable to unfamiliar environments and extreme environmental changes, such as those encountered during evacuations [46]. Frequent changes in evacuation location may have affected the mental and physical health, as well as the overall condition, of these HPNS patients. However, the retrospective nature of this study, which relied on data extracted from nursing records, precludes a definitive determination of this effect. Further detailed investigation is necessary to elucidate these potential effects.

This study had several limitations. First, it used data from a single HPNS provider. The lack of comparison with other HPNS providers limits the generalizability of the findings. Second, to examine factors associated with prolonged evacuation, we established thresholds based on descriptive statistics and data distributions. It should be noted that the 75th percentile value may underestimate or overestimate the number of individuals experiencing prolonged evacuation, depending on the characteristics of the data distribution. The 75th percentile is not an absolute standard but is dependent on the characteristics of the study population. Third, data were collected retrospectively from nursing records. Therefore, it was not possible to assess socioeconomic variables, patients’ condition during the evacuation, or the effect of the evacuation on patients’ mental and physical health. Furthermore, the effects of patient attributes, duration of evacuation, and number of evacuations on participants’ physical and mental health and pre-existing illnesses are unknown. Fourth, the Noto Peninsula is a geographically isolated region with a high aging rate and limited medical and long-term care resources. The earthquake and tsunami caused extensive damage, including collapsed houses, which forced many residents into prolonged evacuation. The timing of the disaster (New Year’s Day) and factors specific to the Noto Peninsula, such as the severe winter climate in the Sea of Japan region, cannot be ruled out as potentially influencing the results. Fifth, because we used a retrospective, chart-based approach, we could not measure systemic factors, such as the quality of inter-shelter communication or the specific referral processes that may have affected evacuee transfers. Future prospective studies should investigate these important logistical and communication aspects of the evacuation process. These findings are based solely on data from HPNS patients affected by the 2024 NPE and are not directly applicable to other countries or regions. To increase the generalizability of the findings, additional studies in other disaster-affected areas, as well as multicenter collaborative research involving larger samples from multiple home-visit nursing agencies, are needed.

## 6. Conclusions

This study investigated the evacuation status and factors associated with prolonged evacuation among patients using HPNS who were affected by the 2024 NPE. Compared with those of previous major disasters, the number of evacuation-related relocations and evacuation duration were exceptionally prolonged in this event. Factors associated with prolonged evacuation in the 2024 NPE were an initial evacuation period of less than 23 days and initial evacuation to a public shelter. Patients exhibiting these factors may have lacked adequate social support, suggesting the need for early interventions tailored to the characteristics of individuals with mental disorders. These findings indicate an urgent need for proactive measures, including establishing systems to identify individuals with special needs, like HPNS users, to ensure that they receive timely and specialized support. Moreover, the findings highlight the importance of disaster preparedness efforts that prospectively identify individuals who require special consideration during non-disaster times and use this information to develop tailored support plans. Furthermore, the geographical vulnerability of the peninsula may have contributed to the prolonged and frequent evacuations, suggesting that preparedness measures for long-term evacuation are more important in peninsular regions than in mainland areas. Additionally, the findings highlight the need to establish support systems that consider geographical characteristics, including securing evacuation sites other than public shelters and conducting disaster preparedness and evacuation simulations specifically for HPNS patients.

## Figures and Tables

**Figure 1 ijerph-22-01688-f001:**
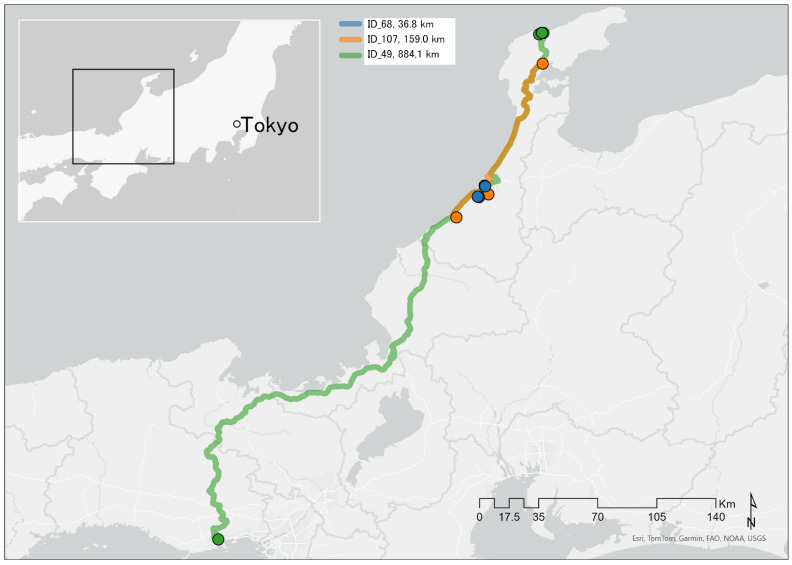
Location of the Noto Peninsula, Japan, and evacuation routes of frequent HPNS evacuees.

**Figure 2 ijerph-22-01688-f002:**
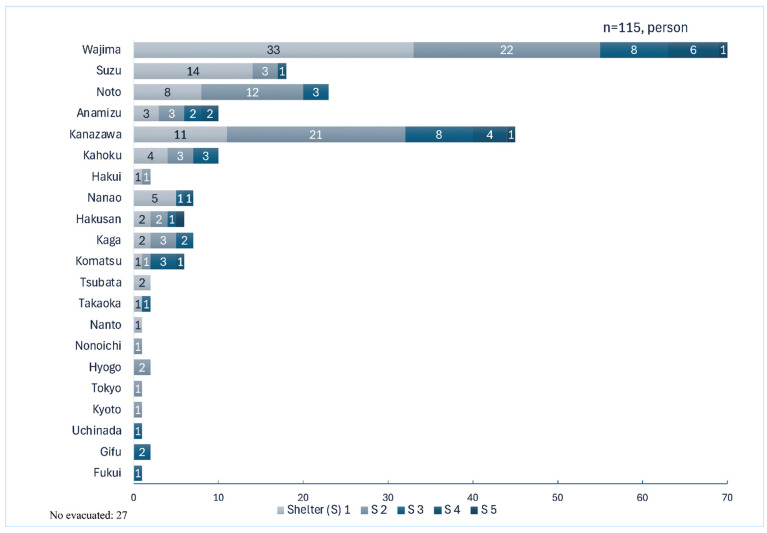
Evacuation time distribution (rounds 1–5).

**Table 1 ijerph-22-01688-t001:** Factors associated with long-term evacuation (*n* = 88).

				Total Number of Evacuation Days				
Items	Category	Total		<75th Percentile Value (<212 Days)		≥75th Percentile Value (≥212 Days)		
		*n*	%	*n*	%	*n*	%	*p*-Value
Participant background								
Sex	Male	38	43.2	20	52.6	18	47.4	1.000
	Female	50	56.8	27	54.0	23	46.0	
Age, years	mean (standard deviation) 53.3 (16.8)							
Age group	<65 years	65	73.9	37	56.9	28	43.1	0.267
	≥65 years	23	26.1	10	43.5	13	56.5	
10-year age group distribution	10s	3	3.4					
	20s	2	2.3					
	30s	17	19.3					
	40s	15	17.0					
	50s	20	22.7					
	60s	12	13.6					
	70s	15	17.0					
	80s	4	4.5					
Main diseases (other than Schizophrenia)								
	Other than Schizophrenia	50	56.8	27	54.0	23	46.0	0.899
	Schizophrenia	38	43.2	20	52.6	18	47.4	
Distribution of main diseases other than schizophrenia	Bipolar disorder	10	11.4					
	Depression	10	11.4					
	Autism spectrum disorder	3	3.4					
	Intellectual disability	3	3.4					
	Alcoholism	5	5.7					
	Autism spectrum disorder	6	6.8					
	Attention-deficit hyperactivity disorder	2	2.3					
	Delusional disorder	2	2.3					
	Adjustment disorder	1	1.1					
	Others	8	9.1					
Evacuation status								
Days of evacuation	Median (IQR)	208 days (192−213)						
Distribution of number of evacuation days by round (1–5)	First evacuation (*n* = 88) Median (IQR)	23 days (12−54)						
	Second evacuation (*n* = 76) Median (IQR)	97 days (22−167)						
	Third evacuation (*n* = 35) Median (IQR)	75 days (36−129)						
	Fourth evacuation (*n* = 17) Median (IQR)	80 days (37−130)						
	Fifth evacuation (*n* = 3) (median (range))	62 days (17−152)						
Number of evacuations	Median (IQR)	2 (2–3)						
Length of stay at the initial shelter	<23 days	41	46.6	14	34.1	27	65.9	<0.001
	≥23 days	47	53.4	33	70.2	14	29.8	
Municipality of residence before the earthquake	Other than Wajima	39	44.3	21	53.8	18	46.2	0.942
	Wajima	49	55.7	56	53.1	23	46.9	
Distribution of municipalities other than Wajima	Suzu	17	19.3					
	Noto	16	18.2					
	Anamizu	6	6.8					
Living situation before the earthquake	Cohabiting	73	83.0	35	47.9	38	52.1	0.023
	Living alone	15	17.0	12	80.0	3	20.0	
Living situation after the earthquake	Cohabiting	30		19	63.3	11	36.7	0.180
	Living alone	58		28	48.3	30	51.7	
Whether the initial shelter was a public shelter	Non-public shelter	37	42.0	28	75.7	9	24.3	<0.001
	Public shelter	51	58.0	19	37.3	32	62.7	
Whether hospitalized during the initial evacuation period	No	84	95.5	44	52.4	40	47.6	0.620 ^a^
	Yes	4	4.5	3	75.0	1	25.0	
Municipality of the initial shelter	Other than Wajima	55	62.5	34	61.8	21	38.2	0.041
	Wajima	33	37.5	13	39.4	20	60.6	
	Other than Suzu	74	84.1	42	56.8	32	43.2	0.148
	Suzu	14	15.9	5	35.7	9	64.3	
	Other than Kanazawa	77	87.5	37	48.1	40	51.9	0.008
	Kanazawa	11	12.5	10	90.9	1	9.1	

^a^ Fisher’s exact test; all other tests were χ^2^ tests. IQR, interquartile range.

**Table 2 ijerph-22-01688-t002:** Predictors of long-term evacuation: binary logistic regression results (*n* = 88).

			95% CI		
Items	Category	OR	Lower Limit	Upper Limit	*p*-Value
Sex	Female/Male	1.24	0.45	3.46	0.679
Age, years	≥65/<65	1.05	0.33	3.29	0.938
Length of stay at the initial shelter	<23/≥23	3.26	1.15	9.26	0.026
Living situation before the earthquake	Cohabiting/Living alone	3.80	0.79	18.23	0.096
Whether the initial shelter was a public shelter	Public shelter/Non-public shelter	4.56	1.49	13.95	0.008
The initial shelter was located in Wajima	Not Wajima/Wajima	1.40	0.45	4.38	0.562
The initial shelter was located in Kanazawa	Not Kanazawa/Kanazawa	5.04	0.53	48.07	0.160

OR: odds ratio; 95% CI: 95% confidence interval.

## Data Availability

The data that support the findings of this study are not publicly available due to them containing information that could compromise research participant privacy.

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
