# Peer review of "A Retrospective Study on Evacuation and Long-Term Displacement Among Home-Visit Psychiatric Nursing Service Users in the Aftermath of the 2024 Noto Peninsula Earthquake"

_ijerph, 2025, doi:10.3390/ijerph22111688_

Round 1

Reviewer 1 Report

Comments and Suggestions for Authors

This is a useful piece of work that focuses on home-visit psychiatric nursing service evacuees during a disaster. It highlights the challenges that give understanding for preparedness interventions in the years to come. The introduction is long but justifiable as it provides readers understanding of the system and geographical limitations.

Line 57- Communications was mentioned by two references as a factor. This could be added in the Discussion part. Could it be that better communication from staff shelter A about the condition in shelter B leads to fewer subsequent transfers? How was the referral made in between shelters -was it through a central agency or just between the shelters?

Line 80- Likewise the authors found lack of coordination as one of the factors (perspective from relief providers ref 16) but I do not see it being discussed in discussion. It is understandable that lack of coordination is hard to measure quantitatively in results but could have been included in discussion.

Fig 1 shows the GIS for three evacuees. Is it possible to do for all 115 stacked in one map? Because it also showed the spread as some wet far as south-east to Tokyo and south-west to Kyoto.

Fig 2. Maybe a change of bar colour (not in shades of grey) will make visualization better.

Line 317 to 326. How about delivery of HPNS service in the new shelters? Did they receive the psychiatric nursing in new sheltes on top of food and lodging? if yes, was it delayed? 

Line 382 in conclusion. Is this a comparison with the general population that does not have psychiatric nursing needs, and therefore, the general population can better adapt to new shelters?

Author Response

Response to the Reviewer 1

We are grateful for the time and effort that you dedicated to providing your valuable feedback on our manuscript. We have carefully considered all your comments and revised the manuscript accordingly.

Line 57- Communications was mentioned by two references as a factor. This could be added in the Discussion part. Could it be that better communication from staff shelter A about the condition in shelter B leads to fewer subsequent transfers? How was the referral made in between shelters -was it through a central agency or just between the shelters?

Response:

We thank the reviewer for this insightful point regarding the role of inter-shelter communication and referral processes. This is indeed a critical factor that could influence evacuee transfers.

However, as our study is a retrospective analysis based solely on information available in medical records, we were unable to obtain detailed data on the specifics of communication between shelters or the nature of the referral system (e.g., whether it was centralized or shelter-to-shelter). Including a discussion on this topic without supporting data would be speculative. Therefore, while we acknowledge the importance of this issue, addressing it is beyond the scope of our current dataset. We have, however, added a note to our limitations section to reflect this.

Line 80- Likewise the authors found lack of coordination as one of the factors (perspective from relief providers ref 16) but I do not see it being discussed in discussion. It is understandable that lack of coordination is hard to measure quantitatively in results but could have been included in discussion.

Response:

Thank you for this observation. You are correct that we mentioned a lack of coordination in the introduction (citing reference 16), and we agree this is a critical aspect of the overall disaster response.

We included this point to provide essential background context on the challenges faced during the 2024 NPE. However, our study is a retrospective analysis based on medical records, which do not contain data that would allow us to quantitatively or qualitatively measure the degree of coordination between different agencies or support providers.

To maintain the scientific rigor of our study and avoid speculation unsupported by our data, we could not include a detailed discussion on this topic. We acknowledge this as a limitation of our research, which we have noted in the manuscript. We believe that investigating the impact of inter-agency coordination would be an important and valuable direction for future research, likely requiring a different study design.

Fig 1 shows the GIS for three evacuees. Is it possible to do for all 115 stacked in one map? Because it also showed the spread as some wet far as south-east to Tokyo and south-west to Kyoto.

Response:

Thank you for this suggestion. While visualizing the routes for all 115 participants on a single map is an interesting idea, it would be technically challenging to implement effectively. Stacking this many line layers would result in severe overlapping, particularly around the primary disaster area, making the individual routes nearly impossible to discern and rendering the map illegible. To analyze the overall geographic spread of evacuees, as you noted, towards Tokyo and Kyoto, a different approach, such as using point layers to represent destinations, would be more appropriate. However, since the specific aim of our GIS analysis was to visualize the complex, multi-stage evacuation routes of the most frequently displaced individuals, rather than the overall distribution of all evacuees, we believe the current approach is better aligned with the study's stated objectives.

Fig 2. Maybe a change of bar colour (not in shades of grey) will make visualization better.

Line 317 to 326. How about delivery of HPNS service in the new shelters? Did they receive the psychiatric nursing in new sheltes on top of food and lodging? if yes, was it delayed?

Response:

Thank you for the suggestion to improve Figure 2. We have revised the figure for better clarity, opting for a high-contrast, monochromatic dark blue scheme instead of multiple colors. This design choice enhances visual clarity and ensures the figure remains fully legible even when printed in black and white.

Line 382 in conclusion. Is this a comparison with the general population that does not have psychiatric nursing needs, and therefore, the general population can better adapt to new shelters?

Response:

Thank you for this question, which allows us to clarify our conclusion.

No, this statement was not intended to be a direct comparison with the general population. Rather, the conclusion is drawn from the specific findings of our study, such as the prolonged evacuation associated with certain factors and the instances of extremely long travel distances. These results suggest that standard evacuation plans may be insufficient when considering the unique difficulties and vulnerabilities faced by this particular patient group.

Reviewer 2 Report

Comments and Suggestions for Authors

The study is interesting; it raises a topic that has been little discussed and explored.

It would have been interesting to delve deeper into the goals of the research, beyond understanding the factors associated with the evacuation.

However, the consequences of the evacuations are not detailed. Once the evacuation was completed, there is no mention of what happened to the people, whether they were able to return to their homes, how long it took, or if they were unable to, etc.

The conclusions suggest that preventive measures should be adopted, given that the area is vulnerable to earthquakes. What should be done? Providing some insight would be a great contribution to the topic.

Author Response

Response to Reviewer 2

We are grateful for the time and effort that you dedicated to providing your valuable feedback on our manuscript. We have carefully considered all your comments and revised the manuscript accordingly.

It would have been interesting to delve deeper into the goals of the research, beyond understanding the factors associated with the evacuation.

However, the consequences of the evacuations are not detailed. Once the evacuation was completed, there is no mention of what happened to the people, whether they were able to return to their homes, how long it took, or if they were unable to, etc.

The conclusions suggest that preventive measures should be adopted, given that the area is vulnerable to earthquakes. What should be done? Providing some insight would be a great contribution to the topic.

Response:

Thank you for your insightful feedback, which has helped us significantly improve the manuscript.

We agree that a deeper discussion of the practical implications of our findings would be a great contribution. In response to your suggestion to provide insight on "What should be done?", we have revised the Conclusion section. We added specific recommendations based on our findings, such as the need for proactive measures to identify and support individuals with special needs and the importance of developing tailored disaster preparedness plans during non-disaster times.

Regarding your comment on the lack of detail about the consequences of the evacuations (e.g., whether people returned home, how long it took), we fully acknowledge this as a limitation. Indeed, this point was also raised by another reviewer, and our position remains consistent: as this was a retrospective study based on available nursing records, we did not have access to systematic data on post-evacuation outcomes. We have sought to clarify this constraint in the Limitations section of our manuscript. While this is a crucial area for investigation, we believe it falls beyond the scope of our current research design.

We appreciate you highlighting these important areas, and we hope our revisions have adequately addressed your concerns.

Reviewer 3 Report

Comments and Suggestions for Authors

The manuscript is valuable and interesting, thank you for the opportunity to review, with a few remarks.

Inconsistency in the conclusion on the '23 days' threshold.
The results state that a shorter stay in the first shelter <23 days  is associated with a higher probability of prolonged evacuation (OR 3.26; 95% CI: 1.15–9.26), but the Conclusion states "23 days or more", which contradicts the model's findings. Please align the text (summary/discussion/conclusion) with the results of the regression.

Dichotomization of continuous variables and thresholds.
Several key characteristics (e.g. length of first stay, age) are dichotomized (e.g. <23 vs. ≥23 days; <65 vs. ≥65 years). Suggest:

  • analyze continuously (or with spline functions),
  • implement sensitivity to different thresholds (e.g. median, tertiles),
  • consider a time-to-event analysis (e.g., Cox, if possible to define "time to completion of evacuation" / stable accommodation). This would reduce the loss of information and the risk of threshold artifacts.

Definition of the outcome 'prolonged evacuation'.
The outcome is defined by the 75th percentile (= ≥212 days), although the median is 208 days; however, in Table 1, the label "≥202 days" (probable typographical error) crept into the header. Please correct 202 → 212 and explain why the 75th percentile, and add sensitivity (e.g. 70th/80th percentile).

There are a few minor typographical and stylistic errors in the text (e.g. duplicate acronym in the image caption; '≥202' instead of '≥212'; "≥65/65" in Table 2). Please proofread and standardize the format (units, decimal points/periods, consistent reference style)

Comments on the Quality of English Language

There are several minor typographical and stylistic errors in the text (e.g., a duplicated acronym in a figure caption; “≥202” instead of “≥212”; “≥65/65” in Table 2). Please copyedit and standardize the formatting (units, decimal separators—comma vs. point—and a consistent reference style).

Author Response

Response to Reviewer 3

We are grateful for the time and effort that you dedicated to providing your valuable feedback on our manuscript. We have carefully considered all your comments and revised the manuscript accordingly.

Inconsistency in the conclusion on the '23 days' threshold.

The results state that a shorter stay in the first shelter <23 days  is associated with a higher probability of prolonged evacuation (OR 3.26; 95% CI: 1.15–9.26), but the Conclusion states "23 days or more", which contradicts the model's findings. Please align the text (summary/discussion/conclusion) with the results of the regression.

Response:

Thank you for pointing out the discrepancy in our figures. We have reviewed the document and corrected the abstract and conclusion accordingly.

Dichotomization of continuous variables and thresholds.
Several key characteristics (e.g. length of first stay, age) are dichotomized (e.g. <23 vs. ≥23 days; <65 vs. ≥65 years). Suggest:

  • analyze continuously (or with spline functions),
  • implement sensitivity to different thresholds (e.g. median, tertiles),
  • consider a time-to-event analysis (e.g., Cox, if possible to define "time to completion of evacuation" / stable accommodation). This would reduce the loss of information and the risk of threshold artifacts.

Response:

We thank the reviewer for these insightful suggestions regarding the statistical analysis.

Our primary objective for this study was to identify clear and practical factors associated with the binary outcome of prolonged evacuation among psychiatric home-visit service users. The decision to dichotomize continuous variables, such as the length of initial stay (<23 days) and age (<65 years), was a deliberate choice aimed at enhancing the clinical interpretability and applicability of our findings for practitioners in the field. We believe that providing specific thresholds offers more straightforward guidance for disaster response planning compared to interpreting the more complex per-unit odds ratios from continuous variables.

While we agree that sensitivity and time-to-event analyses are powerful methods that can reduce information loss, we consider them to be beyond the scope of our current retrospective study. However, we recognize the value of these approaches and will certainly consider these recommendations as a reference for our future research on this topic.

Definition of the outcome 'prolonged evacuation'.
The outcome is defined by the 75th percentile (= ≥212 days), although the median is 208 days; however, in Table 1, the label "≥202 days" (probable typographical error) crept into the header. Please correct 202 → 212 and explain why the 75th percentile, and add sensitivity (e.g. 70th/80th percentile).

Response:

Thank you for your valuable and constructive comments.

First, regarding the typographical error in Table 1, we have corrected the value from “202 days” to “212 days” as you pointed out.

Concerning the rationale for defining prolonged evacuation using the 75th percentile, we chose this threshold because the median total number of evacuation days was already significantly prolonged at 208 days. We aimed to specifically identify the subgroup that experienced an “extremely prolonged” duration of displacement even within this already long-term context. We have added this justification to the Methods section of the manuscript.

Regarding the suggestion for sensitivity analysis (e.g., exploring different thresholds), we appreciate this insightful recommendation. While we fully acknowledge the importance of sensitivity analysis, our study primarily relies on retrospective data, derived from medical records, which inherently present limitations in terms of granularity and detailed precision of evacuation duration records. Therefore, conducting a comprehensive sensitivity analysis using multiple, finely-tuned thresholds was deemed beyond the scope of the current work, particularly given its primary objective of providing clinically interpretable findings for practical application. However, we consider this suggestion to be a very valuable perspective for future research, and we will certainly consider it for subsequent studies

There are a few minor typographical and stylistic errors in the text (e.g. duplicate acronym in the image caption; ‘≥202’ instead of ‘≥212’; “≥65/65” in Table 2). Please proofread and standardize the format (units, decimal points/periods, consistent reference style)

Comments on the Quality of English Language

There are several minor typographical and stylistic errors in the text (e.g., a duplicated acronym in a figure caption; “≥202” instead of “≥212”; “≥65/65” in Table 2). Please copyedit and standardize the formatting (units, decimal separators—comma vs. point—and a consistent reference style).

Response:

We appreciate your pointing out the typographical and stylistic errors in our manuscript. We have corrected all the specific issues you raised and thoroughly proofread the remainder of the text for consistency (e.g., units, punctuation, and referencing style). Furthermore, the revised manuscript has been reviewed and approved by a native English speaker.